# Military Service and Offending Behaviors of Emerging Adults: A Conceptual Review

Christopher Salvatore [1,*] and Travis Taniguchi [2]

1   Department of Justice Studies, Montclair State University, Montclair, NJ 07043, USA
2   National Police Foundation, 2550 South Clark Street, Suite 1130, Arlington, VA 22202, USA;
    ttaniguchi@policefoundation.org
*   Correspondence: salvatorec@montclair.edu

**Abstract:** Focusing on the United States, this paper examines the impact of military service for the cohort of individuals that have experienced the social factors that characterize emerging adulthood as a unique stage in the life course. We argue that military service, as a turning point, may act differently in contemporary times compared to findings from past research. This difference is driven by changes in military service, the draft versus volunteer military service, and the prevalence of emerging adulthood. As a background, we describe emerging adulthood, examine how emerging adulthood relates to crime and deviance, explore the impact of military life on young adults, provide an overview of the demographics of military service, discuss the influence and outcomes of military life on young adults, and explore existing research linking military service and deviant and criminal behavior. We develop a theoretical model of the relationship between military service and emerging adulthood and explore the impact on criminological theory and policy.

**Keywords:** military service; crime; deviance; emerging adulthood; youth crime; social bonds; turning points





## 1. Introduction

Life course and developmental perspectives play a prominent role in explaining deviance, crime, and desistance from deviance and crime. Life course perspectives study individual patterns of behavior over time. A key goal of this work is to understand how some social bonds and turning points (e.g., getting married or having children) act to reduce or prevent anti-social behavior such as criminal offending and deviance from social norms (Laub and Sampson 2003). Research has identified key turning points in people's lives, such as getting married, having children, and attaining career-level employment. Military service has also been identified as a turning point in the lives of many who serve (Laub and Sampson 2003); the present literature is inconclusive regarding the influences of military service on decreasing or preventing behavior, however, some studies have found military services is linked to changes in criminal and deviant behavior, timing of marriage and parenthood, and differences in family formation and family structure (Laub and Sampson 2003; Kelty et al. 2010; Wright et al. 2005). However, existing work linking military service with life changes is limited by a lack of theoretical framework relevant to the current configuration of military service or the social realities of youth in contemporary society. For many, military service is a decisive turning point away from crime (Laub and Sampson 2003), although some scholars have found military service is associated with increased drug use (Wright et al. 2005).

The age-graded theory of informal social control, proposed by Sampson and Laub (1993), has frequently been employed to explain the role of military service on criminal offending. Sampson and Laub's theory focuses on the role of prosocial bonds to conventional society. These bonds are fluid; they vary throughout, and in response to, life stages and events. Social bonds may exert influence on an individual's inclination towards deviant

and criminal behavior. As such, an individual on a path towards a criminal career may build attachments to society that act to reduce criminal offending.

More generally, employment has consistently been found to be a crucial turning point in the lives of offenders and one of the activities that generates the most robust attachments to society. As a result, it has been found to be strongly associated with reduced offending (Salvatore and Taniguchi 2012). Military service, in particular, has been seen as a highly respectable profession, which may provide those with criminal or deviant tendencies a prosocial outlet (Gloyd and Leal 2018). However, much of the literature examining the role of military service as a turning point (e.g., Laub and Sampson 2003) has not considered the potential influence of emerging adulthood which is a stage of the life course first identified by Arnett in 1994.

Emerging adulthood is the prolonged period between childhood and adulthood that is characterized by exploration and experimentation among several domains. Emerging adulthood and changes to military service may make past studies of military service less relevant to current young adults for two reasons. First, contemporary military service has social and structural differences compared to military service in the past. Military service is now entirely volunteer; this voluntary nature may change the socio demographic composition of those that serve. Second, prior research has not explored the impact of service on individuals experiencing emerging adulthood. As a relatively recently identified life course stage, there are reasons to believe that military service may act differently for people in this protracted time between adolescence and adulthood.

This paper is a conceptual review that explores the role of military service as a turning point for those who also experience the life phase of emerging adulthood. We begin by providing an overview of emerging adulthood, followed by a discussion of crime and deviance during this stage of life. Next, we describe the role of military service on criminal and deviant behavior and explore the research that has examined the influence of military service on the behavior of emerging adults. Finally, we place military service in a theoretical framework to better understand its role in the behaviors of emerging adults and describe the implications for theory and policy.

## 2. What Is Emerging Adulthood?

Since the 1960s, the road to adulthood in the United States, and other high-income countries, has undergone considerable changes (Arnett 2015). In particular, the period between adolescence and adulthood has been redefined and is now considerably longer than in the recent past. Traditional markers of reaching "adulthood", such as getting married or having children are being delayed. This has led to a more protracted transition to adulthood relative to generations born before the 1960s (Arnett 2015; Cote 2000). This transitional phase, the period between adolescence and adulthood, has been labeled as emerging adulthood (Arnett 2015). Arnett originally conceptualized emerging adulthood as the period when people are between the ages of 18 and 25. More recently this age range was expanded to include the late 20s (Arnett 2015). Studies examining emerging adulthood, such as Salvatore and Taniguchi (2012), have argued for the need to expand the age range of emerging adulthood even further.

A key aspect of emerging adulthood is that it is a period of exploration and experimentation across life domains such as romantic and sexual relationships, education, and employment (Arnett 2015). Arnett (2015) argued cultural and social changes occurring since the 1960s have influenced behaviors at both group- and individual-levels. He identified four critical areas of change that have led to the genesis of emerging adulthood as a distinct stage of the life course. First, the economy changed from primarily manufacturing, which did not require high levels of education, to more information and service-based economy, which necessitates higher levels of education. Youth in prior generations were able to live a middle-class lifestyle with high school education. More recent generations are required to pursue additional education to attain the same type of lifestyle.

Second, women have increased access to, and expectations of, employment and education. This has increased the likelihood of women postponing marriage and parenthood, while they pursued educational and professional goals. Third, cultural norms have embraced greater sexual freedoms and tolerance of sexual diversity. This has led to the normalization of premarital sex and increased acceptance of homosexuality. Finally, the rise of the youth movement has advanced the narrative that adult life is an unfortunate, and often unwanted, change from the carefree nature of adolescence. The trappings of adult life (e.g., home ownership, long-term single employer career) are something to defer as long as possible in exchange for continued exploration of education, relationships, and experiences. As a result, many who in the past would have viewed themselves as young adults instead chose to frame their choices in terms more consistent with a state of prolonged adolescence (Arnett 2015).

Together, these changes have led to emerging adulthood as a distinct stage of the life course. It is characterized by instability, change, exploration, and examination in a variety of life domains. Concurrently, emerging adulthood is characterized by weakened social bonds to family and society. These weakened bonds may reduce internal self-control that would act to limit criminal and deviance behaviors (Salvatore and Taniguchi 2012).

*Emerging Adulthood and Criminal and Deviant Behaviors*

As emerging adulthood is a stage characterized by experimentation and instability, the social bonds developed by marriage and parenting may not exist, while those like employment and education may be insufficient. Parents, teachers, and employers may not be able to exert the quantity or quality of social bonds to prevent deviant and criminal behaviors. Scholars such as Moffitt et al. (2002) and Piquero et al. (2002) found emerging adulthood to be a period marked by considerable criminality. More recent studies, such as Marcus (2009), Salvatore and Taniguchi (2012), and Maynard et al. (2015), confirmed emerging adulthood as a period of active criminal offending.

These findings are bolstered by official statistics on the age of offenders known to the criminal justice system. Data from the Uniform Crime Report (UCR), indicate that individuals age 18 to 29 have higher rates of arrest for most violent and non-violent crimes compared to other age groups (Uniform Crime Report 2017). Further evidence of the criminogenic nature of emerging adulthood can be found by comparing population levels with arrest levels. The population aged 20 to 29 make up approximately 13% of the U.S. (U.S. Census) but emerging adults (aged 18 to 29) account for 42% of arrests (Uniform Crime Report 2017), and 19% of inmates in federal prison (Federal Bureau of Prisons 2020). The pattern persists at the state level. For example, in Massachusetts, emerging adults made up about 10% of the population in 2013, but accounted for 29% of those arrested, and 20% of those incarcerated by the state (Perket and Chester 2017).

Emerging adulthood is also characterized by increased rates of substance use. The Monitoring the Future study is a multi-wave, nationally representative sample following high school graduates through adulthood. Results from this study suggests that individuals in emerging adulthood (aged approximately 19 to 30) have high rates of marijuana, alcohol, LSD, MDMA (ecstasy), and nicotine use in the past 30 days (Schulenberg et al. 2018). Other behaviors such as dangerous driving have also been found to occur at high rates among people in the emerging adulthood age range; the National Highway Traffic Safety Administration found most vehicle collisions are concentrated among people age 21 to 25 (National Highway Traffic Safety Administration NHTSA). Scholars such as Ferguson (2003) and Hingson (2010), have hypothesized that younger drives take more risks than older drivers, including speeding and driving while intoxicated.

In sum, studies have found emerging adulthood a stage of the life course where crime, deviance, and risky behaviors are common. Researchers have attempted to explain these behaviors through the experimental nature of emerging adulthood and lack of comprehension of the long-term costs associated with these behaviors.

### 3. The Impact of Military Service on the Development of Young Adults

Military service may facilitate the formation of key social bonds that act as a turning point and contribute to the transition to adulthood. Kelty et al. (2010) have noted military service influences family formation, marriage, parenthood, and family structure. Research in this area has some inconsistencies. Similar to emerging adulthood, some research suggests that military service delays traditional markers of the transition to adulthood like marriage and parenting, acting to put life on 'pause' while serving (Shanahan 2000). Others, however, have found the stable income and housing of military service may encourage marriage (Lundquist and Xu 2014). This suggests that the role of military service as a turning point during emerging adulthood may be conditional on demographic factors such as gender, age, race/ethnicity, socioeconomic status, and sexual orientation (Kelty et al. 2010).

Evidence also suggests that the role of military service as a social bond generator and turning point may vary over time. Once the transition to an all-volunteer force occurred there was a marked increase in marriage, as well as couples serving in the military. As of 2018, 12.9% of those serving across all branches of the military were married (United States Department of Defense 2018). In more contemporary times, military service members were somewhat more likely to be married relative to their same aged peers when they started service (Kelty et al. 2010). Others have found some may enter military service single but tend to marry younger. According to Lundquist and Xu (2014) this may be the result of financial circumstances and quicker deployments which have become a bigger challenge in recent years; being married and having a family can make deployment easier. Further, Lundquist and Xu (2014) argue the culture of the military itself influences those serving to get married because it acts as a tool of retention.

Military service provides steady employment, medical benefits for the family, and a path of upward mobility for those who may have only a high school education, allowing for class mobility (Lundquist and Xu 2014; Wang et al. 2012). The financial stability and fringe benefits of military service provide favorable conditions for marriage. The need for support and family during deployment may further encourage those serving to get married (Hogan and Furst-Seifert 2010; Lundquist and Xu 2014). In other words, unlike the World War II generation, those serving in the modern era may find military life encourages the turning point of marriage.

As discussed above, scholars such as Arnett (2015) have found parenthood to be a key turning point to adulthood. Recent information from the United States Department of Defense found 30.9% of those who were married to a civilian and actively serving had children. This number drops to 2.5% when both parents are in military service (United States Department of Defense 2018). Among unmarried service members, 4% of those actively serving have children (United States Department of Defense 2018).

While deployments and absenteeism from day-to-day life and family events may pose challenges for parenting, many of the policies and benefits of military service create an environment where parenting may be encouraged. This may facilitate a quicker transition to parenthood relative to those of the same age not serving in the military. For example, the military provides free medical care, schooling, well-organized recreational opportunities for children and families, and housing based on family size (Kelty et al. 2010). Relative to their same aged peers, those in the military become first time parents sooner, with 21 to 25 being the most frequent age range for females in the military having children (United States Department of Defense 2018), relative to the mean age of 26.9 for women not serving in the military (Martin et al. 2019).

Together, existing research suggests that military life may have many benefits for young adults in making the transition to adulthood. Studies indicate that service in the military in the modern era may be more likely to achieve a turning point and build social bonds sooner than peers not serving in the military. Historically, the military has been dominated by males. More recently, the demographics of those serving in the military has shifted (Barroso 2019). In the next section we explore these changing demographics.

*Changing Demographics of Military Service*

Overall number of people in the military has decreased. In recent years the size of the military has decreased to approximately 1.34 million in 2017 from 1.46 million in 2010, a considerable decrease from the peak of 2 million in 1990 (Barroso 2019). Across the four branches of the military (i.e., Army, Navy, Air Force, and Marines) the Army has the largest number of personnel with approximately 500,000 service members (Reynolds and Shendruk 2018).

For prior generations, such as the men involved with Laub and Sampson (2003) research, the United States military was largely composed of White males. However, the demographics of those serving in the military has changed over the last 50 years (Barroso 2019). Most military service members are still White, but the representation of minority service members continues to grow (Barroso 2019). In 2017, 57% of those in service were white, followed by Hispanics and Blacks at 16% each, respectively, Asians at 4%, and 6% identified as other or unknown (Barroso 2019). In regard to gender, females were 16% of those serving in the military in 2017, a marked increase from the 9% of females in 1980, and a significant jump from 1% of females in the military in 1970 (Barroso 2019). Women have also increased representation among commissioned officers. According to Reynolds and Shendruk (2018) women represented 18% of officers in 2016 which was a substantial increase from the 8% they represented in 1973. In 2016, the top five states in regard to recruitment were Georgia, Florida, South Carolina, Virginia, and Arizona (relative to their military service eligible populations).

The research presented so far has suggested that military service may promote social bonds and create a turning point in the lives of those who serve. By providing conditions amenable to marriage and family, service members may find themselves reaching turning points that encourage prosocial behavior and discourage crime and deviance (Arnett 2015; Salvatore 2018). However, in the following section, we provide an alternative perspective on the role of the military on individuals, namely that military service may encourage or facilitate antisocial behavior including crime and drug abuse.

## 4. The Influence of Military Service on Criminal and Deviant Behavior

The relationship between military service and criminal and deviant behavior has received intense study. Some scholars have argued that military service is a decisive turning point away from youth offending and leads to positive life outcomes (Elder and Shanahan 2006). Others find military service increases criminal offending post service (Bouffard 2005). The divergent results in this research may be driven by the historical and social context around military service.

Studies examining military service around the time periods involving World War II and the Korean War have typically found better outcomes for military service than studies that have been conducted on more recent military service (Salvatore and Taniguchi 2012). Many of these studies have considerable challenges: they have relied on relatively small samples, with little demographic diversity (e.g., Laub and Sampson 2003). More recent studies such as Salvatore and Taniguchi (2012), which used data from the National Longitudinal Study of Adolescent and Adult Health, had a small sample of those actively or formerly serving in the military, reducing the generalizability of the findings.

### 4.1. Crime and Deviance during and after Active Military Service

In order to fully understand the relationship between military service and crime, a distinction must be made between criminal and deviant behavior occurring while in the military versus behaviors that occur post-military service. Crimes involving military service members can come in a number of forms: (1) actions that violate military law and subject to court martial, (2) actions that violate civilian state or local laws, or (3) actions that violate foreign criminal laws while stationed abroad. Jurisdictions may overlap; offenders may be subject to both civilian and military law. We explore each of these relationships to

inform our theoretical perspective on the role of military service on people experiencing emerging adulthood.

Court Martials are convened to try military service members for violations of the Uniform Code of Military Justice (UCMJ; Snedeker 1949). The UCMJ covers a wide range of offenses from minor infractions to the most serious of offenses and includes service member activity that occurs both in relation to military service and from their private non-military activity (Monroe 1942). Each branch of the military publishes monthly court martial statistics. This information provides an insight into the number and types of offenses occurring. For example, in March of 2020, the U.S. Marine Corps reported 21 court martial, for a variety of crimes including assault and battery of a spouse, attempted sexual assault and abuse of child, violation of general order, extra marital sexual conduct, and drunk and disorderly conduct (United States Marine Corps 2020). In May of 2019, the United States Air Force reported there were 32 Court Martials for a variety of crimes ranging from sexual assault, lewd behavior, distributing child pornography, possession and distribution of controlled substances, and indecent visual recording (United States Air Force 2019).

Other types of criminal activity committed by active service military members may fall under non-military jurisdictions. Crimes may be committed by military service members, but subject to the jurisdiction of civilian law. This may include crimes committed within the United States or in foreign lands where the service member is stationed. The scope of crimes committed by active-duty military is difficult to establish. In aggregate, research findings are mixed on the impact of military personnel on crime in places they are stationed. For example, Allen and Flynn (2013) found that greater numbers of deployed US troops increased crime in the host country only under some conditions and for some types of crimes.

Turning to prior service, studies have found both positive and negative impacts for those who served in the military. For example, a 1954 study by Bennett explored the relationship between involvement in crime of non-veterans compared to veterans of World War II and the Korean War (Bennett 1954). They found veterans were less likely to be sent to a federal prison compared to non-veterans. The protective value of being a veteran was strongest for violent crime outcomes. In their landmark 2003 study, Laub and Sampson argued that military service "knifes off" prior (delinquent and criminal) experiences, taking former delinquents out of negative peer groups and communities, and serves as a positive transition.

However, upon closer examination they found that military service was not a decisive turning point for everyone. For some individuals, military service failed to act as a turning point and offending continued after service suggesting that the protective factor of military service may be conditional on other situations or person-level characteristics. Military service, similar to substance abuse disorders or mental health disorders, may increase criminal offending by disconnecting social relationships and roles (Sampson and Laub 1996). Other studies, such as Bryant (1979), suggest that military service may give those who served the training and expertise in combat and weapons which may be applied to conflicts outside the military.

### 4.2. The Influence of Time Period on the Link between Military Service and Crime and Deviance

Research suggests that the link between military service and crime varies based on the time period of service. Vietnam was, at best, not a protective factor against offending, or at worst, was associated with *greater* offending. Bouffard (2003) found that, after holding juvenile police contacts constant, those who served in Vietnam were as likely to have come in contact with the police as individuals who had not served in the military. Others like Wright et al. (2005) found that Vietnam veterans were more likely to report higher rates of substance abuse compared to those who did not serve in Vietnam or had no military service. In a similar manner, more recently, Snowden et al. (2017) found that military service was associated with higher levels of crime. However, like Sampson and Laub, Snowden et al.

found this relationship was inconsistent. For Snowden et al., the results were driven by a subgroup of military service members that had poor fit with military service and served for only a few years.

Wright et al. (2005) also identified the importance of understanding how social class impacts the military-offending connection. They found that individuals with lower socioeconomic status, who had pre-established delinquency patterns, had a higher chance of serving in Vietnam, and their service in Vietnam was associated with increased drug use and drug offending. While Wright et al. (2005) employed a rigorous methodology, using 8 waves of data to accurately study behavioral trends such as drug use, there were several noteworthy limitations. For example, their sample was relatively homogeneous. A more heterogeneous sample by including individuals after the age of 26 would better reflect demographics of the military (and general society).

In late 1970, the Gates Commission recommended the end of the use of the draft during times of war and shifted to an All-Volunteer Force (AVF) (Warner and Asch 2001). Culp et al. (2013) explored the impact of this change by using data from Surveys of Inmates of State and Federal Correctional Facilities and the Current Population Surveys from 1985 to 2004. Their results suggest that, in aggregate, military service was not predictive of incarceration after controlling for social integration and demographics. However, once results were disaggregated, those serving during the AVF era had a higher likelihood of being incarcerated compared to those of the draft era. While a comprehensive and detailed study, Culp et al.'s study may fail to capture the full scope of those experiencing the shift from the draft to AVF and military services influence on offending. Data from surveys of the general population, such as the General Social Survey (GSS), could be used to capture the influence of military service on offending for those who were not incarcerated.

A core concern for researchers is attempting to understand why different generations (e.g., World War II and Vietnam eras), and pre-AVF and AVF military services, have different influences on offending behavior. MacLean and Elder (2007) stated economics may play a key factor; those serving in the pre-AVF period had greater economic opportunities compared to those in the AVF era. More broadly, the modern day AVF military may be meaningfully different from the pre-AVF era. A large portion of studies (e.g., Laub and Sampson 2003) examined the influence of military service on offending involved individuals serving in the pre-AVF military. The modern era AVF military is composed of disproportionately minority racial/ethnic groups (relative to their numbers in the population) from middle-class backgrounds whereas the pre-AVF era drew across a broader social and economic background (Parker et al. 2017; Reynolds and Shendruk 2018). Military personnel drawn across socio demographic lines could have led to greater inclusion of individuals less predisposed to offending or reduced inclusion of those with prior delinquent offending.

Additionally, greater economic opportunities provided to draft-era Veterans relative to those leaving the military in the AVF Era, may have resulted in improved outcomes. Those in lower socioeconomic status groups in the pre-AVF era were able to take advantage of the benefits offered through the G.I. Bill including educational, training, and housing opportunities coinciding with the suburban 'boom' and strong post-World War II economy. In contrast, those in the post-AVF era face challenges with the need for increased education and skills in an economy driven by information and service, rather than manufacturing.

Other scholars have pointed to variations in the public response to military service as a source of differentiation. Those who served in the military during World War II and Korea came back to parades, stability, and were considered by many to be heroes. In contrast, those who served in the Vietnam era returned to a country riddled with conflict and struggling with changing norms and values. The Vietnam War was protested and those serving in the military were less likely to experience accolades from the public upon returning from service (Kidder 1978). More recently the public has returned to a more celebratory view of veterans. According to a 2012 Pew Foundation Study, 9 out of 10 Americans report they feel proud of those who have served in Iraq and Afghanistan wars,

a marked shift from the protests that greeted Vietnam Veterans when they returned home (Cohen and Funk 2011).

Studies of veterans from the post-9/11 era have found many combat veterans reported difficulties with family, have a higher level of irritability and anger, and increased symptoms of post-traumatic stress (Parker 2011). In addition to general challenges with military service, research has found that sexual assault in the military is prevalent throughout the military, including in areas where combat is occurring (Street et al. 2009). A recent Department of Defense report found that in fiscal year 2017, 6769 reports of sexual assault were made where service members were either victims or subjects of a criminal investigation. It should be noted, the 2017 figures reflect a significant increase of 9.7 percent increase from the 6172 reports made in the prior year (Ferdinando 2018).

## 5. Emerging Adulthood and Military Service

Research examining the first cohorts of people to experience emerging adulthood and serve in the military has produced mixed results. The number of veterans incarcerated in state and federal prisons, and local jails decreased from over 200,000 in 2004 to approximately 182,000 in 2012 (Bronson et al. 2015), suggesting those who have experienced emerging adulthood may be less likely to offend. During the years between 2001 and 2012, veterans from Operation Iraqi Freedom, Operation Enduring Freedom, and Operation New Dawn, a large portion of whom were born post 1960, accounted for 25% of veterans in jail and 13% in state and federal prison (Bronson et al. 2015). The rate of incarceration for veterans was approximately the same in 2011–2012, as it was in 1998 (Bronson et al. 2015), suggesting that military service members that have experienced emerging adulthood may not be more criminogenic relative to other cohorts.

Some evidence suggests that cohorts who have experienced emerging adulthood may be more likely to commit specific types of crime. For example, Bronson et al. (2015) found that incarcerated veterans were most likely to be incarcerated for violent sexual assault, followed by other violent crimes, property crimes, drug offense, and DUI/DWI (Bronson et al. 2015). A key challenge with many studies of the influence of the military on offending behaviors is the use of data from prisons and jails, which may miss those who commit crimes, but are not arrested and incarcerated.

As noted above, a large portion of the literature exploring the influence of military service has been conducted on samples from World War II, Korean, and Vietnam eras. People serving during this time period were not subject to the societal changes characterizing emerging adulthood. Little research has explored the influence of military service during the time period involving emerging adulthood. In the remainder of this section, we discuss theoretical models to better understand the relationship between military service, emerging adulthood, and criminal and deviant behavior. The authors based the selection of these theoretical models based on the framework and direction prior studies have provided (e.g., Salvatore 2018) regarding the influence of emerging adulthood on criminal and deviant behavior.

First, the military acts as a total institution (Goffman 1961) which controls all facets of life for those in service. Day to day routines, including activities, meals, and sleep, are standardized; these are carried out on a rigid schedule for which members must comply. These strict routines are meant to prepare people for hostile events and environments and promote cohesion and sense of group within the military (Sun et al. 2007). This structure, combined with a reward system that promotes compliance and solidarity, socializes members of the military to be conformists, bonded to the group and institution, leaving little acceptance of those who venture outside the bounds by engaging in crime and deviance. Further, the military provides vocational and technical training, which may help individuals establish stronger social bonds (Elder 1999). As such, those in the military 'learn' to follow rules and guidelines and seek to conform, not wanting to risk rejection by engaging in behaviors that would risk membership in the group.

There is some evidence to support the value of the military as a total institution on reducing offending behaviors. For example, one of the men in Laub and Sampson (2003) study cited that the military offered food, clothing, structure, and discipline for socioeconomically disadvantaged men (p. 132). Similar to reform schools, the military offered a highly structured environment that provides for all the needs of the individual and promotes a sense of group identity, discouraging deviance and offending.

On the other hand, some scholars have argued that military service acts to teach people to solve their problems using aggression, which may lead to problems in civilian life including criminal offending (Hakeem 1946). Cohen et al. (1992) found service in Vietnam lead to adverse life events including being socially disenfranchised and discriminated against, which others have found to be correlated with offending and substance use (Wright et al. 2005).

As a total institution, military service may offer emerging adults, a group characterized by experimentation and change, increased stability and focus. Structuring day-to-day activities such as waking up, work, eating, and recreation may provide emerging adults with an environment in which their experimental nature and identity exploration may be constrained. This increased external control may reduce tendencies to engage in substance use, crime, and other behaviors that act as a form of identity exploration. Identity exploration is an often cited (Salvatore 2018) characteristic of emerging adulthood and one of the primary reasons emerging adults engage in deviant and criminal behaviors.

The next theoretical explanation is Sampson and Laub's age-graded theory of informal social control (Sampson and Laub 1993). Drawing from the life course literature, Sampson and Laub argue that family is the most important form of social control during early stages of life. Direct control via parental supervision, and indirect control via family attachment, help to prevent deviance and delinquency. Delinquency occurs where discipline is ineffective, parental monitoring is insufficient, or social bonds with the family are attenuated. A key factor in Sampson and Laub's Theory is the notion of cumulative continuity, described in the context where crime and responses to crime are tied to individuals offending trajectory (Sampson and Laub 1993).

Sampson and Laub's framework suggest that offenders will establish new bonds and attachments during adulthood; these bonds may act as turning points, leading them to a more prosocial trajectory. In their 2003 study, Laub and Sampson found many men who had served in World War II indicated military service was an effective turning point. Their period of service cut off past criminal trajectories and put them on a prosocial path. However, the military was not an effective turning point for all men in the study. For some, military service was ineffective due to the nature of their service. For example, one study participant stated that military service in the army made things worse due to exposure to combat. He speculated that if his service had occurred during peacetime, he would have had a better outcome (p. 176). This suggests that exposure to combat, or the stress of being in an active military, may be a factor that negatively influences the ability of the military to act as a positive turning point.

As with prior generations, service in the military by people experiencing emerging adulthood may benefit from the ability of the military to provide them a fresh start, free of criminogenic communities and peer groups. The military may give emerging adults a turning point towards adult life, steering them away from deviance and crime, while fostering social bonds that can act to inhibit crime and deviance. Further, once separated from the military, people in emerging adulthood may have marketable skills and post-service benefits that include college education or vocational skills training. Attachment to career produces prosocial bonds which studies have found help encourage desistance from crime and deviance (e.g., Salvatore and Taniguchi 2012).

The third theoretical explanation that may explain the influence of military service on emerging adults is the emerging adulthood gap. This perspective was proposed by Salvatore (2018) and is grounded in Moffitt (1993) developmental taxonomy of two offender types: the adolescent limited (AL) and life-course persistent (LCP) offender. AL offenders

typically engage in low-level crimes like underage drinking and shoplifting. The AL offender often starts engaging in delinquency in their teens and stops by the end of high school. LCP offenders, by contrast, demonstrate antisocial behavior much earlier in the life course and engage in both lower-level delinquency and more serious crimes like robbery. Unlike the AL offender, LCP offenders continue to offend past their teens.

As discussed above, scholars have found emerging adulthood to be an active period of offending. Salvatore (2018) emerging adulthood gap, provides a conceptual and theoretical link between emerging adulthood and offending. As Moffitt proposed a maturity gap during adolescence, Salvatore proposed a similar gap during emerging adulthood. During the emerging adulthood gap, individuals increase experimentation with sexuality, drugs, and alcohol, and criminal behaviors. Scholars like Arnett (2015) have argued the experimental nature of emerging adulthood is facilitated by decreased level of informal social controls. Salvatore (2018) emerging adulthood gap argues that, due to delays in turning points and lack of social bonds in emerging adulthood, individuals fall into another 'gap' during which offending behaviors, substance abuse, and other risky and dangerous behaviors may be prevalent.

Consistent with studies of previous generations, military service may act as a strong bond and turning point for emerging adults. Individuals that join the military right from high school may be able to avoid the emerging adulthood gap, and thereby reduce antisocial behaviors that are more common during emerging adulthood. For people already engaged in antisocial behaviors due to the emerging adulthood gap, military service may serve as a turning point and produce social bonds. Similar to prior generations, the military may remove individuals from delinquent neighborhoods and peer groups and place them in a highly structured and controlled environment that reduces the likelihood of engaging in crime and deviance.

Although not empirically tested, the emerging adulthood gap theory incorporates aspects of the age-graded theory of informal social control, control theory, and Moffitt's developmental taxonomy. It provides a conceptual framework for future studies to examine the role of the military as a turning point and social bond that influences the offending behaviors of emerging adults. The three theoretical explanations presented here provide a framework to examine the potential influence of military service on emerging adults. Each of the three perspectives provides a sociological context through which we can explain how military service can influence the offending behaviors of people that experience emerging adulthood.

## 6. Implications for Practice and Theory

The influence of military service has been explored in the social science literature for quite some time, but the bulk of this research has focused on the generations who served in World War II, the Korean War, and the Vietnam War. Although informative, there is reason to study the influence of military service on criminal offending in more contemporary times. Critically, existing research has not provided a comprehensive assessment of the influence of military service on the criminal offending behaviors of youth that are, or have, experienced the more recent development of emerging adulthood.

A considerable number of studies have linked military service with criminal behaviors and substance use. The direction of these relationships, however, appear to be at least partially contingent on the timeframe of the study. Older research has generally found military service associated with reduced offending. More recent studies have found military service associated with increased substance abuse and sexual assaults. The reasons for these differences in findings are worth further study and may at least in part be explained by changes in social structure that are impacting people's transition from adolescence to adulthood.

Some studies suggest that military service may act as a key social bond that inhibits criminal behaviors in emerging adults. The military acts as a total institution and provides for all needs of its members. Military service is characterized by a strong culture that

encourages conformity and dissuades behaviors and actions that violate organizational norms. Studies performed with prior generations found that military service could effectively cut off individuals from prior criminal peers and communities, provide structure, and facilitate pro-social career growth through skills training and post-service college education.

A more limited body of research has studied military service on individuals that have encountered the recently identified developmental phase of emerging adulthood. This research suggests military service may still have an impact on offending behaviors and substance use. Theories such as social control, the age-graded theory of social control, and the emerging adulthood gap help to clarify the theoretical connection between military service, emerging adulthood, and the process by which military service "works" to influence the behavior of emerging adults.

Turning to policy implications, at-risk individuals, or those who are engaged in antisocial behaviors during emerging adulthood, may be directed towards military service or other kinds of institutions that provide structural elements analogous to the military. As with previous generations, military service may provide a way out of communities with inadequate social controls, a lack of prosocial role models, and few opportunities for employment. Those already involved in crime, such as those in Laub and Sampson (2003) sample who had a history of juvenile delinquency, may find the military offers structure and discipline, as well as a fresh start, away from negative influences of the past. However, a challenge, especially with those who may already be on a criminogenic path, is that the military could offer opportunities to engage in crime like sexual assault. As such, care should be taken to create a culture within the military that does not tolerate sexual assaults. It is also possible some of the characteristics of military service, such as skills training and education assistance, could be ported out of the military and become normalized as a resource that is available to everyone.

*Future Research*

There is a need for continued research exploring the influence of military service for individuals involved in emerging adulthood. Emerging adulthood is a relatively recent development in life course research. Therefore, issues associated with the long-term influences of military service are still unknown among the cohort of individuals that have experienced, or will experience, emerging adulthood. Future research should explore basic questions regarding the role of military service and emerging adulthood. For example, the demographic characteristics of people serving in the military is changing over time. It is unknown if the impact of military service will be consistent across race, gender, and socioeconomic status.

Using the lens of the theories discussed above, future studies should test the influence of military service on the offending behaviors of emerging adults. Scholars may utilize these theoretical perspectives to clarify if the current generations experience military service as a net benefit. Existing research tends to suggest that military service operates as a beneficial social bond and culture; it provides opportunities to those who enlist to acquire new skill sets and move away from delinquency of their youth. Nevertheless, other research suggests that military service may have the opposite effect in that it may encourage offending behaviors and create physical and emotional trauma that may lead to substance abuse problems or offending.

We know relatively little about undetected crime and deviance committed by current and former military service members. Most existing studies tend to use samples of individuals who are incarcerated. This focus, however, ignores the fact that many crimes are not reported to the police and even those reported to the police tend to go unsolved (Schmalleger 2018). Unreported crimes are a persistent problem for criminal justice researchers. Exploring this issue further may require qualitative approaches, such as conducting in-depth interviews, to access information on unreported or undetected criminality. Using more in-depth interviews, instead of surveys, could allow for greater details

about the role of marriage, parenting, and the influence of military service on offending behaviors. Further, through open-ended qualitative interviews, research participants could be probed for additional issues such as the influence of media images, popular culture, socioeconomic status, and other factors that may influence behaviors. More critical than determining if military service acts as a positive or negative turning point is determining what factors associated with military service act on individuals to increase or decrease the likelihood of involvement in crime or other deviant behavior. Understanding these factors can be used to develop additional research that could improve the value of military service for service members. Additionally, understanding these characteristics could lead to more effective crime and delinquency prevention programs that could be implemented in a non-military setting. Some characteristics of military service, such as skills training or strong team focus, can be implemented in other settings that may not have the same ethical implications as military service.

Emerging adulthood is a relatively new stage of the life course and the bulk of longitudinal studies exploring the influence of military service have been conducted with prior generations that did not experience prolonged emerging adulthood. There is a need for scholars to examine the impact of military service on emerging adults retrospectively, this may be the only method through which the long-term influence of the military on emerging adults may be discovered. Prospective studies should be conducted that would allow follow-up of cohorts as they go through different stages of the life course. It is of interest to conduct comparison studies of those in military service with non-military peers. Further, within the military cohort, it would be possible to differentiate between those more in line with emerging adulthood and others with the traditional model of young adulthood.

Further, reintegration into a non-military setting and long-term career trajectories post-military has not been studied on people that have experienced emerging adulthood. For many, military service occurs during the period between emerging adulthood and prolonged adolescence. In theory, when people leave military service, they are older, more mature, have skills that facilitate entry into stable careers, and have more pro-social bonds that turn them away from crime or deviance. Alternatively, separation from military service may result in loss of peer groups, financial instability, and the need to adapt to the loss of total institution controls. Additional research is needed to understand these characteristics of post-military service for current generations.

Finally, the limited existing research on military services and emerging adulthood has tended to focus on the United States and other high-income nations. There is a need to expand beyond this limited scope to better understand how military service impacts emerging adulthood and the transition into adulthood.

## 7. Conclusions

This paper examines the critical concepts related to the influence of military service on antisocial and criminal behaviors of people in emerging adulthood. While military service has been a component of prior studies examining emerging adulthood, there has yet to be a comprehensive examination of the role of military service as a key social bond or turning point during emerging adulthood that may influence offending behaviors of emerging adults. Recent studies such as Gloyd and Leal (2018) have found potential benefits of military service for emerging adults. Nevertheless, additional research is needed to explore the influence of military service on the offending behaviors of emerging adults. This is especially important given the prevalence of minority and lower-income individuals in military service. These same sociodemographic characteristics are, unfortunately, also over-represented throughout the criminal justice system.

Better understanding of how military service impacts individuals in the phase of emerging adulthood may lead to better social programs that are able to support the changing needs of youth transitioning to adulthood. There is a need to disentangle the main characteristics of military service (e.g., strong social controls versus job and career

training) that contribute to military service as a potential turning point. Further, the influence of military service on criminal behavior cannot be analyzed in isolation. Changes in stages of the life course such as the evolution of emerging adulthood may alter how military service 'works' in the lives of those who serve. In addition, the effect of military service may depend on the crossover influence of a variety of different variables such as sociodemographic characteristics, prior delinquency, or criminality before entering service.

**Author Contributions:** Conceptualization—C.S.; Writing—C.S. and T.T.; Original draft preparation—C.S. and T.T.; Writing Review and Editing—C.S. and T.T. All authors have read and agreed to the published version of the manuscript.

**Funding:** This research received no external funding.

**Conflicts of Interest:** The authors declare no conflict of interest.

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
