# Peer review of "Military Service and Offending Behaviors of Emerging Adults: A Conceptual Review"

_socsci, doi:10.3390/socsci10020049_

Round 1

Reviewer 1 Report

The paper presents a review of literature on antisocial and criminal behaviours of 
people in emerging adulthood, the impact of military service on the development of Young Adults 
and its influence on Criminal and Deviant Behavior during and after the service in the US.

The paper clearly shows the limits of existing research, especially in relation to the current situation.

It also suggests that further research, in particular, if carried out with a more refined sampling and supported by more qualitative methodology may help improve the understanding of the phenomenon and provide suggestions to improve social programs designed to support emerging adulthood

The paper is clearly written and readable

Author Response

The authors thank Reviewer 1 for their comments.

Reviewer 2 Report

The manuscript presented by the authors raises an interesting issue, the influence of military service on offending behavior in emerging adulthood. However, the paper presents a major limitation, the lack of a specific section describing the methodology used to conduct the scoping review. In my comments to the authors, I develop this and other suggestions for improvement.

1. Major comments:

1.1. Title. The authors should clearly establish the type of review they have carried out. The title specifies "Research Review", while the main body of the text defines the study as a "scoping review". The design of the study should be clearly specified in the title ("Scoping Review").

1.2. Methodology and results. The authors should describe in detail the methodology used to conduct the scoping review. The manuscript should be written following the PRISMA statement for scoping reviews (http://www.prisma-statement.org/Extensions/ScopingReviews; Tricco, AC, Lillie, E, Zarin, W, O'Brien, KK, Colquhoun, H, Levac, D, Moher, D, Peters, MD, Horsley, T, Weeks, L, Hempel, S, et al. PRISMA extension for scoping reviews (PRISMA-ScR): checklist and explanation. Ann Intern Med. 2018,169(7):467-473. doi:10.7326/M18-0850). The methodology section should include eligibility criteria, information sources, search, selection of sources of evidence, data charting process, data items, and synthesis of results (data charted). The results section should include the selection of sources of evidence, characteristics of sources of evidence, results of individual sources of evidence, and synthesis of results. This is probably the most important weakness that authors should address.

1.3. Introduction (page 1, lines 22-24). The authors assume from the beginning of the paper a positive influence of military service and emerging adulthood on the prevention of antisocial behavior (criminal offending and deviance from social norms). The text further specifies that the results of the available research are inconclusive in this regard. Considering the limitations of available studies on this topic, the authors should be more cautious in their statements and define their key goal without pointing out the direction of the influence of military service on offending behaviors.

1.4. Emerging Adulthood. It is evident that social and cultural patterns have changed, and this allows us to refer, in general terms, to a new stage of life (emerging adulthood). Nevertheless, it is possible that in current society there are individuals who experience a traditional transition (more direct) between adolescence and adulthood and others who enter this new and longer phase of emerging adulthood. To assume that all people in society are currently experiencing emerging adulthood is risky. This assumption has implications for research aimed at determining the differential effect of military service on emerging adulthood. Comparing current results with studies from past generations does not guarantee control of other variables that may be biasing the outcome. Therefore, it is interesting that future research controls for the "life stage" variable (emerging adulthood yes/no) and, in this way, to make comparisons between coexisting groups.

1.5. Military service. Military service temporarily coincides with the transition phase from adolescence to adulthood, although the characteristics of this vital stage may be different depending on the generation. One of the research challenges is determining the influence that military service alone has on an individual's behavior. Its effect is expected to be mediated by other variables. The authors repeatedly define military service as a decisive turning point away from crime and highlight its role in establishing social bonds. However, any occupation can play a significant role in terms of criminality and social bonds. In many cases, the behavior observed may be more determined by the stage of change (transition between adolescence and adulthood) than by the specific military context.

1.6. If possible, it would be interesting for the authors to provide data on emerging adulthood crime rates in the non-military population.

1.7. To highlight the possible positive effect of military service, the authors barely go into detail about its possible negative effect on the life experience and behavior of emerging adults. The rigidity and structure provided by military service are a double-edged sword. It can be positive, but it can also be very stressful. Military service can foster an attitude of service and volunteerism, but it also separates people from their natural social contexts. In short, I perceive an imbalance in the analysis of the possible effects of military service, although it is true that the authors mention both directions.

1.8. If the authors follow the PRISMA protocol, in the results section it is recommended that they specify the number of articles included in the review that support a positive influence of military service and those that evidence a negative one.

1.9. Conclusion. It should be emphasized that the influence of military service on criminal behavior cannot be analyzed in isolation. Its effect depends on the crossover influence of different variables (sociodemographic characteristics, experience in military service, the transition from adolescence to adulthood, etc.).

2. Minor comments

2.1. Page 5 (lines 180-187). Need a reference.

2.2. Page 6 (line 213). This section of the article should be structured in subsections for clarity.

2.3. Page 7 (lines 287-288). This sentence seems out of context.

2.4. Page 9 (lines 344-351). This text could be relocated later when describing the lines of future research (section 6.1.).

2.5. Page 13 (lines 541-543). In addition to conducting retrospective studies, it would also be interesting to conduct prospective studies that would allow follow-up of cohorts over the years. It would be interesting to compare people in military service with non-military peers. Also, within the military cohort, it would be possible to differentiate between profiles that are more in line with emerging adulthood and others with the traditional model.

Author Response

Response to Reviewer 2

We thank the reviewer for their comments. In response to the edits requested please see below:

  1. Major comments:

1.1. Title. The authors should clearly establish the type of review they have carried out. The title specifies "Research Review", while the main body of the text defines the study as a "scoping review". The design of the study should be clearly specified in the title ("Scoping Review").

Authors Response: The authors thank the reviewer for their comment and agree the title should be amended. The goal of the paper is to conceptually tie military service and emerging adulthood together (which has not been done before), as such the title has been amended to reflect this goal.

1.2. Methodology and results. The authors should describe in detail the methodology used to conduct the scoping review. The manuscript should be written following the PRISMA statement for scoping reviews (http://www.prisma-statement.org/Extensions/ScopingReviews; Tricco, AC, Lillie, E, Zarin, W, O'Brien, KK, Colquhoun, H, Levac, D, Moher, D, Peters, MD, Horsley, T, Weeks, L, Hempel, S, et al. PRISMA extension for scoping reviews (PRISMA-ScR): checklist and explanation. Ann Intern Med. 2018,169(7):467-473. doi:10.7326/M18-0850). The methodology section should include eligibility criteria, information sources, search, selection of sources of evidence, data charting process, data items, and synthesis of results (data charted). The results section should include the selection of sources of evidence, characteristics of sources of evidence, results of individual sources of evidence, and synthesis of results. This is probably the most important weakness that authors should address.

Author Response: The authors appreciate the guidance provided in this comment. As mentioned above the goal of the present paper is a conceptual review to connect the idea of emerging adulthood and military service. The authors are planning a more formal scoping review, followed by an empirical paper, as part of their future research agenda. Text within the narrative has been amended to ‘conceptual review’ on line 60. This paper represents a first step in an overall agenda, and given that emerging adulthood is a fairly new area of inquiry, and military service has not been examined within the context of emerging adulthood , the authors feel it is an important and necessary one needed to lay the foundation for the rest of the research agenda in this area.

1.3. Introduction (page 1, lines 22-24). The authors assume from the beginning of the paper a positive influence of military service and emerging adulthood on the prevention of antisocial behavior (criminal offending and deviance from social norms). The text further specifies that the results of the available research are inconclusive in this regard. Considering the limitations of available studies on this topic, the authors should be more cautious in their statements and define their key goal without pointing out the direction of the influence of military service on offending behaviors.

Authors Response: The authors appreciate the feedback regarding this section. The text has been amended to reflect that the current literature is inconclusive, though some studies do suggest military service may reduce criminal and antisocial behavior for emerging adults serving in the military (see lines 28-30).

1.4. Emerging Adulthood. It is evident that social and cultural patterns have changed, and this allows us to refer, in general terms, to a new stage of life (emerging adulthood). Nevertheless, it is possible that in current society there are individuals who experience a traditional transition (more direct) between adolescence and adulthood and others who enter this new and longer phase of emerging adulthood. To assume that all people in society are currently experiencing emerging adulthood is risky. This assumption has implications for research aimed at determining the differential effect of military service on emerging adulthood. Comparing current results with studies from past generations does not guarantee control of other variables that may be biasing the outcome. Therefore, it is interesting that future research controls for the "life stage" variable (emerging adulthood yes/no) and, in this way, to make comparisons between coexisting groups.

Authors Response: The authors appreciate the insights being offered by the reviewer. There are certainly those who may be experiencing the more traditional ‘young adulthood’ of the past. Since research in emerging adulthood is relatively new (compared to more general scholarship in adolescences and young adulthood), future studies should incorporate indicators of being in ‘emerging adulthood’ relative to ‘young adulthood,’ something the authors will keep in mind as they move forward with their research agenda in this area.

1.5. Military service. Military service temporarily coincides with the transition phase from adolescence to adulthood, although the characteristics of this vital stage may be different depending on the generation. One of the research challenges is determining the influence that military service alone has on an individual's behavior. Its effect is expected to be mediated by other variables. The authors repeatedly define military service as a decisive turning point away from crime and highlight its role in establishing social bonds. However, any occupation can play a significant role in terms of criminality and social bonds. In many cases, the behavior observed may be more determined by the stage of change (transition between adolescence and adulthood) than by the specific military context.

Authors Response: The authors thank the reviewer for their comment. We agree that any occupation may play the role of a positive turning point—though there could be factors such as antisocial peers within the context of the occupation

1.6. If possible, it would be interesting for the authors to provide data on emerging adulthood crime rates in the non-military population.

Authors Response: The authors agree that the inclusion of crime rates in non-military emerging adults would be an interesting and important section for the paper. Unfortunately, most studies looking at populations of emerging adults or crime/deviance in emerging adults simply focus on those in the age group (rather than specific markers of emerging adulthood itself, such as delayed transitions). As such the authors felt the lack of a cohesive narrative in this area would limit the utility of discussion of these data.

1.7. To highlight the possible positive effect of military service, the authors barely go into detail about its possible negative effect on the life experience and behavior of emerging adults. The rigidity and structure provided by military service are a double-edged sword. It can be positive, but it can also be very stressful. Military service can foster an attitude of service and volunteerism, but it also separates people from their natural social contexts. In short, I perceive an imbalance in the analysis of the possible effects of military service, although it is true that the authors mention both directions.

Authors Response: The authors appreciate the comment provide above. One of the challenges in working within the topic area of emerging adulthood and military service is there is little direct empirical studies examining the relationship between emerging adulthood and military service as it relates to criminal offending or drug use. As stated previously, this is part of the rationale of the authors for pursing this avenue of inquiry—to help establish a conceptual framework in the present paper, provide a more detailed examination of existing literature in a follow up paper, then empirically examine the relationship through qualitative and quantitative studies. The information provided in section 4 does examine the potential negative influence of military service on those serving.

1.8. If the authors follow the PRISMA protocol, in the results section it is recommended that they specify the number of articles included in the review that support a positive influence of military service and those that evidence a negative one.

Authors Response: Per the response to comments 1 and 2, the authors have edited the paper to clarify it is a conceptual review.

1.9. Conclusion. It should be emphasized that the influence of military service on criminal behavior cannot be analyzed in isolation. Its effect depends on the crossover influence of different variables (sociodemographic characteristics, experience in military service, the transition from adolescence to adulthood, etc.).

  Authors Response: The authors appreciate this comment and have added text to the conclusion to address these points (lines 595-599).

  1. Minor comments

2.1. Page 5 (lines 180-187). Need a reference.

Authors Response: Per the reviewers note, we have added a reference in this section.

2.2. Page 6 (line 213). This section of the article should be structured in subsections for clarity.

Authors Response: The authors appreciate this helpful suggestion and have added subsections to help clarify the discussion in this section.

2.3. Page 7 (lines 287-288). This sentence seems out of context.

Authors Response: The authors agree with the original placement of this text was out of context and moved this text to an above paragraph (lines 284-288).

2.4. Page 9 (lines 344-351). This text could be relocated later when describing the lines of future research (section 6.1.).

Authors Response: The authors agree with the reviewer and have moved this text to section 6.1, as suggested.

2.5. Page 13 (lines 541-543). In addition to conducting retrospective studies, it would also be interesting to conduct prospective studies that would allow follow-up of cohorts over the years. It would be interesting to compare people in military service with non-military peers. Also, within the military cohort, it would be possible to differentiate between profiles that are more in line with emerging adulthood and others with the traditional model.

Author Response: The authors appreciate the feedback on these areas and have added them to the discussion (lines 564-568)

Round 2

Reviewer 2 Report

I thank the authors for accepting my comments and appreciate their efforts to reflect them in the manuscript.